# GraphTorque: Torque-Driven Rewiring Graph Neural Network

## Abstract

Graph Neural Networks (GNNs) have emerged as powerful tools for learning from graph-structured data, leveraging message passing to diffuse information and update node representations. However, most efforts have suggested that native interactions encoded in the graph may not be friendly for this process, motivating the development of graph rewiring methods. In this work, we propose a torque-driven hierarchical rewiring strategy, inspired by the notion of torque in classical mechanics, dynamically modulating message passing to enhance representation learning in heterophilous and homophilous graphs. Specifically, we define the torque by treating the feature distance as a "lever arm vector" and the neighbor feature as a "force vector" weighted by the homophily ratio disparity between node pairs. We use the metric to hierarchically reconfigure each layer's receptive field by automatically pruning high-torque edges and adding low-torque links based on a Bernoulli-guided learnable sampling process, suppressing the impact of irrelevant information and boosting pertinent signals during message passing. Extensive evaluations on benchmark datasets show that the proposed approach surpasses state-of-the-art rewiring methods on both heterophilous and homophilous graphs.

## 1 Introduction

Graph-structured data composed of vertices and edges encode entities and their relationships. Graph neural networks (GNNs) have emerged as a powerful framework for processing such data, with widespread applications in biomolecular modelling Gligorijević et al. (2021); Xia et al. (2023), recommendation systems Chen et al. (2024); Anand & Maurya (2025) and beyond Jiang et al. (2023); Liu et al. (2025); Huang et al. (2025). At the heart of GNNs lies message passing, which iteratively propagates and aggregates information along edges to enrich node representations. Therefore, the graph structure not only encodes entity interactions but also critically determines model performance Zhang et al. (2020); Yang et al. (2023); Qian et al. (2024).

In practice, however, raw graphs frequently harbour spurious or missing links arising from noise or sampling artefacts, compromising their effectiveness as substrates for message propagation. In response, recent work has devised diverse graph rewiring strategies that selectively remove or add edges to optimize message passing and boost predictive accuracy Xue et al. (2023); Bi et al. (2024); Liang et al. (2025). Such dynamic topology adjustment is crucial not only for mitigating spurious connections but also for addressing heterophily, where nodes with dissimilar labels or features tend to be connected Yang et al. (2021); Zheng et al. (2023); Li et al. (2025). In such scenarios, homophily-based GNNs can be misled by abundant heterophilous connections, yielding entangled representations and degraded classification accuracy.

One of the core challenges in graph rewiring is quantifying the significance of edges on message passing. A key factor in this process is the similarity between node pairs, often measured using the Euclidean distance, a widely used metric for assessing similarity. In general, the greater the distance between nodes, the weaker their interaction strength, and the less useful information can be transmitted, as supported by previous studies that employed node similarity as a proxy for edge weights Wang et al. (2020); Zhou et al. (2024). To intuitively observe this, we simulate adversarial attacks by injecting adversarial edges into raw graphs and visualize the distance distribution of the edges, enabling us to examine whether adversarial and original edges exhibit distinct distributional patterns. As shown in Fig. 1(a)–(d), the distribution trends in both homophilous datasets (Cora

and Pubmed) and heterophilous ones (Wisconsin and Texas) consistently reveal that adversarial edges (in red) tend to connect node pairs with larger feature distances. This suggests that adversarial attacks preferentially create long-range links so that they disrupt message passing at their target nodes. Furthermore, we observe that normal edges in heterophilous datasets also exhibit a distribution skewed toward larger distances, more markedly than in homophilous datasets. This arises because heterophilous graphs contain a substantially higher proportion of heterophilous edges, which typically connect node pairs with low similarity. Given that a minority of long-range neighbors can convey crucial information while nearby neighbors may propagate misleading signals, the feature quality of neighboring nodes should be considered another key factor in assessing edge significance.

This brings to mind the concept of **_Torque_** in classical mechanics, which is mathematically defined as the cross product of a lever arm (the position vector from the axis of rotation to the point of force application) and a force. Recently, torque has found applications in fields such as biology Tang et al. (2023); Dzhimak et al. (2022); Drobotenko et al. (2025) and spintronics Kovarik et al. (2024); Camarasa-Gómez et al. (2024). Heuristically, we extend this concept to graphs by treating the distance vector between nodes as the lever arm and the feature vector of a neighboring node as the force. Their product yields a graph torque, which measures an edge's negative impact: higher torque value flags greater interference. To our knowledge, this is the first work to integrate a physics-inspired torque into graph rewiring, enabling an interference-aware message passing.

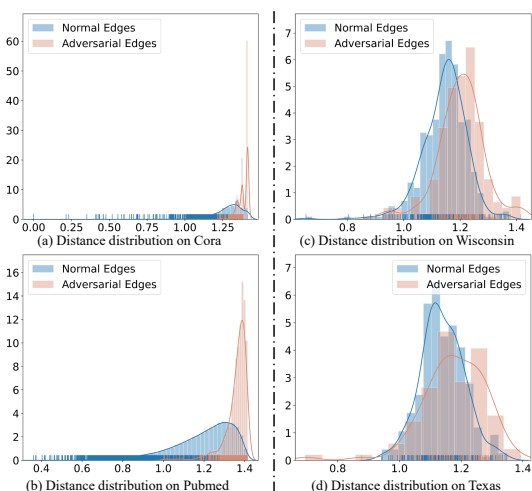

Figure 1: Density distributions of distances for normal vs. adversarial edges on homophilous graphs (a) Cora and (b) Pubmed and heterophilous graphs (c) Wisconsin and (d) Texas.

Specifically, we devise a Torque-driven Hierarchical Rewiring strategy (THR) for GNNs, which dynamically refines message passing to excel in both homophilous and heterophilous graph. In THR, each edge is assigned a torque value that quantifies its interference strength, with larger torques indicating less reliable connections. Torque is defined by treating the difference between node representations as a "lever arm vector", which emphasizes long-range or heterophilous links. Meanwhile, the neighbor feature is regarded as a "force vector" weighted by the disparity in the node-level homophily ratios. This disparity captures the difference in their local label homophily, which has been theoretically shown to jointly influence the expressive power of GNNs, alongside feature distance. Leveraging this torque value, THR hierarchically reconfigures each layer's receptive field via automatically removing undesirable edges that degrade performance and introducing low-torque significant connections through a Bernoulli-guided learnable sampling process. This enables interference-resistant and importance-aware propagation. This rewiring is performed end-to-end, where message passing operates on the continuously updated graph, while the evolving node representations enhance torque computation.

**Contributions:** 1) To the best of our knowledge, we are the first to apply the concept of torque from physics to graph rewiring, resulting in THR, which enhances GNNs' resilience to both homophily and heterophily. 2) We propose a hierarchical rewiring strategy that adaptively determines each layer's receptive field by automatically pruning undesirable connections and learnably sampling significant edges. 3) Comprehensive experiments indicate that THR improves the performance of various GNNs and outperforms existing state-of-the-art rewiring strategies.

## 2 RELATED WORK

Standard message passing in GNNs, which aggregates information from local neighbourhoods, struggles to capture long-range dependencies. A common remedy is to stack multiple layers to

expand the receptive field Wu et al. (2019); Chen et al. (2020); Xu et al. (2025), but this often leads to fundamental issues such as over-smoothing and over-squashing. To overcome these bottlenecks, graph rewiring techniques have recently emerged as an effective strategy for restructuring connectivity and enhancing information flow. For example, Expander GNNs and ExPhormer perform graph rewiring by merging multi-hop neighbourhoods or injecting virtual nodes Gabrielsson et al. (2023); Shirzad et al. (2023). Karhadkar et al. (2022) adds edges based on spectral expansion to mitigate over-smoothing and over-squashing, while degree-preserving local edge-flip algorithms are proposed by Banerjee et al. (2022). Saber & Salehi-Abari (2025) introduces a causal method to assess the impact of graph rewiring on over-squashing, enabling selective rewiring. Topping et al. (2022); Di Giovanni et al. (2023) analyze the root causes of over-squashing, demonstrating that both spatial and spectral rewiring can effectively counteract this bottleneck.

Moreover, Bo et al. (2021) highlights the challenge posed by heterophilous edges, where the aggregation of dissimilar node signals can entangle representations and cause misclassifications. To alleviate the impact of such undesirable connections, several methods employ graph rewiring to improve representation learning. For instance, Bi et al. (2024) compares the neighbourhood feature and label distributions between node pairs, pruning heterophilous edges while introducing homophilous ones. Bose et al. (2025) leverages autoencoders to derive reweighted similarity coefficients, thereby strengthening graph homophily. Other approaches, such as Yan et al. (2022); Luan et al. (2022); Liang et al. (2023), adopt signed message propagation, assigning positive weights to homophilous links and negative weights to heterophilous ones. This enables differentiated updates on heterophilous graphs, amplifying similarity among homophilous nodes while suppressing misleading signals from heterophilous nodes. However, Liang et al. (2024) shows that although a single-hop signed adjacency matrix aids in separating class features, extending this to multi-hop propagation often degrades performance.

We draw inspiration from torque in physics to develop a novel rewiring mechanism that hierarchically eliminates undesirable connections and incorporates task-relevant edges. By dynamically reshaping the receptive field during training, our method enhances the discriminative power of GNNs on both homophilous and heterophilous graphs.

## 3 PRELIMINARIES

### 3.1 NOTATIONS

Let us define an undirected graph dataset as $\mathcal{G} = (V, \mathcal{E})$, comprising $N$ nodes $\{v_i \in V\}_{i=1}^N$ and $K$ edges $\{e_k \doteq \langle i,j \rangle \in \mathcal{E}\}_{k=1}^K$, where each edge $k$ encodes a connection between nodes $v_i$ and $v_j$. We denote the adjacency matrix by $\mathbf{A} \in \{0,1\}^{N \times N}$, where $A_{\langle i,j \rangle} = 1$ iff nodes $v_i$ and $v_j$ are connected, 0 otherwise. Furthermore, $\widehat{\mathbf{A}} = \mathbf{A} + \mathbf{I}$ indicates $\mathbf{A}$ with added self-loops, and $\widetilde{\mathbf{A}} = \widehat{\mathbf{D}}^{-1/2}\widehat{\mathbf{A}}\widehat{\mathbf{D}}^{-1/2}$ denotes the symmetrically normalized adjacency matrix with $\widehat{D}_{\langle i,i \rangle} = \sum_{i=1}^N \widehat{A}_{\langle i,j \rangle}$. Each node is associated with a feature vector, and we denote the node feature matrix by $\mathbf{X} \in \mathbb{R}^{N \times d}$, where the $i$-th row, $\mathbf{x}_i \in \mathbb{R}^d$, represents the $d$-dimensional features of node $v_i$. Among $N$ nodes, $N_{lab}$ nodes are labeled, with ground-truth labels encoded in a matrix $\mathbf{Y} \in \mathbb{R}^{N_{lab} \times c}$, where each row $\mathbf{y}_i$ is a one-hot vector indicating the class label among $c$ categories.

### 3.2 MESSAGE PASSING

Consider a graph with adjacency matrix $\mathbf{A}$ and node feature matrix $\mathbf{X}$. Message passing in a GNN proceeds by iteratively propagating and aggregating neighborhood information as

$$\mathbf{h}_i^{(l+1)} = \text{Upd}\big(\mathbf{h}_i^{(l)}, \sum_{v_j \in \mathcal{N}_i} \text{Agg}(\mathbf{h}_j^{(l)}, A_{\langle i,j \rangle})\big), \tag{1}$$

where $\mathbf{h}_i^{(0)} = \mathbf{x}_i$, and $\mathbf{h}_i^{(l+1)} \in \mathbb{R}^m$ is the representation of node $v_i$ in the $(l+1)$-th layer. $\text{Agg}(\cdot)$ computes the incoming message from a neighbor $v_j$, and $\text{Upd}(\cdot)$ updates the representation of node $v_i$. Rather than relying on the raw adjacency matrix $\mathbf{A}$, most GNNs adopt a modified propagation operator $\mathcal{A}$. For example, GAT Velickovic et al. (2018) replaces each non-zero entry of $\mathbf{A}$ with a learned attention coefficient that depends on the representations of the corresponding node pair.

### 3.3 Node-level Homophily and Heterophily

For a set of nodes with labels, the homophily ratio of each node quantifies the tendency of the node to share the same label as its neighbors Pei et al. (2020); Luan et al. (2022). Considering a node $v_i$ and its set of neighbors $\mathcal{N}_i$, the homophily ratio $h_i^+$ of $v_i$ is defined as: $h_i^+ = \frac{|\{\mathbf{y}_i = \mathbf{y}_j | v_j \in \mathcal{N}_i\}|}{|\mathcal{N}_i|}$. The value of $h_i^+$ lies in the range $[0, 1]$, where values closer to 1 indicate a higher degree of homophily (or lower heterophily), while values nearer to 0 signify the opposite. To quantify the homophily of the entire graph $\mathcal{G}$, we compute the average homophily across all nodes: $\mathcal{H}(\mathcal{G}) = \frac{\sum_{i=1}^N h_i^+}{N}$.

## 4 Methodology

In this section, we propose a novel graph rewiring strategy that unfolds in three key stages: (i) computing edge torques, (ii) rewiring propagation matrix, and (iii) adjusting message passing. The full algorithmic pseudocode is provided in Appendix A.

### 4.1 Derive Graph Torque

In classical mechanics, **torque** is defined as the vector cross product of a force and its lever arm:

$$\mathbf{T} = \mathbf{r} \times \mathbf{F}, \quad |\mathbf{T}| = |\mathbf{r}||\mathbf{F}| \sin\theta, \tag{2}$$

where $\mathbf{r}$ denotes the position vector, $\mathbf{F}$ indicates the force vector, and $\theta$ is the angle between them. The magnitude of torque in classical mechanics governs an object's tendency to rotate under an applied force. Similarly, in GNNs, the strength of node interactions, which depends on factors such as node similarity, determines how the central node is updated by its neighbors. Just as torque in mechanics results from the interaction between force and lever arm, the graph structure and node features determine the propagation and aggregation of information, ultimately optimizing node representations. Specifically, the torque on a graph can be conceptualized by treating the displacement vector $\mathbf{D}_{\langle i,j \rangle}$ between a central node $v_i$ and its neighbor $v_j$ as the "lever arm", while the features of the neighbor $\mathbf{x}_j$ act as the applied "force". However, the contribution of this force varies across different central nodes. Recent studies Mao et al. (2023); Huang et al. (2025) have demonstrated that the generalization of GNNs is influenced by two key factors: the proximity of aggregated features and the disparity in homophily ratios, with smaller values yielding better generalization. Inspired by this, we introduce the homophily ratio disparity term $E_{\langle i,j \rangle}$ to modulate the force, thereby capturing the heterogeneous influences of neighboring nodes and unifying these two factors within the torque framework to enhance model performance on test data.

Mathematically, for an edge $k$ connecting nodes $v_i$ and $v_j$, we define the corresponding torque as follow,

$$\mathbf{T}_{e_k} = \mathbf{D}_{\langle i,j \rangle} \times E_{\langle i,j \rangle} \mathbf{x}_j. \tag{3}$$

Its magnitude, denoted $T_{e_k}$, quantifies the impact of message passing along edge $e_k$ on node $v_i$. The value increases with larger distance or higher homophily ratio disparity, with edges maximizing both factors yielding the greatest torque value that represents the highest priority for graph rewiring. A central goal is therefore to provide a principled definition of the displacement vector $\mathbf{D}_{\langle i,j \rangle}$ and the homophily ratio disparity $E_{\langle i,j \rangle}$ in Eq. 3, followed by a detailed description of their construction.

**Metric 1: Displacement Vector.** To mitigate the effect of noise in raw graphs and features, we compute the displacement vector $\mathbf{D}_{\langle i,j \rangle}$ using optimized node representations and obtain the distance value $D_{\langle i,j \rangle}$ as follows

$$\mathbf{D}_{\langle i,j \rangle} = \mathbf{h}_i - \mathbf{h}_j, D_{\langle i,j \rangle} = \|\mathbf{h}_i - \mathbf{h}_j\|_2, \tag{4}$$

where $\mathbf{h}_i = \text{gCov}(\mathbf{x}_i, \mathbf{A}; \mathbf{\Theta})$[1] denotes the representation of $v_i$ obtained via a graph convolution operator "gCov" parameterized by $\mathbf{\Theta}$.

**Metric 2: Homophily Ratio Disparity.** Considering that recent studies emphasize the importance of capturing the homophily ratio disparity in addressing heterophilous graphs, we weight the neighboring features using this disparity, incorporating it into the torque formulation. To estimate node-level homophily, it is essential to annotate the labels of neighboring nodes around a given node. Given

---

[1] "gCov" can be instantiated with any standard GNN layer, such as GCN, GPRGNN, or APPNP.

the scarcity of labeled data, we leverage the model's outputs to generate pseudo-labels for unlabeled nodes, with their accuracy improving as the model undergoes progressive optimization. Formally, $E_{\langle i,j \rangle}$ is computed by

$$E_{\langle i,j \rangle} = |h_i^+ - h_j^+|, \ h_i^+ = \frac{|\{v_j | \hat{\mathbf{y}}_i = \hat{\mathbf{y}}_j, v_j \in \mathcal{N}_i\}|}{|\mathcal{N}_i|}. \tag{5}$$

Here, $\hat{\mathbf{y}}_i$ denotes the ground-truth label for labeled nodes or the pseudo-label for unlabeled nodes. Finally, the torque value of edge $e_k$ is computed by:

$$T_{e_k} = \|\mathbf{D}_{\langle i,j \rangle} \times (E_{\langle i,j \rangle}\mathbf{h}_j)\|_2 = \sqrt{D_{\langle i,j \rangle}^2 \cdot (E_{\langle i,j \rangle}\|\mathbf{h}_j\|_2)^2 - (E_{\langle i,j \rangle}\mathbf{D}_{\langle i,j \rangle} \cdot \mathbf{h}_j)^2}.^2 \tag{6}$$

This formulation captures the combined effects of distance and disparity, facilitating a physics-inspired approach to graph rewiring.

## 4.2 ADJUST MESSAGE PASSING

**Edge-removal High-order Rewiring.** Herein, we propose an automated threshold learning mechanism that identifies the optimal number of edges to prune by pinpointing the largest successive torque gap. Specifically, we first rank all $K$ edges in descending order of their torque values to form a torque-sorted list (TSL), denoting its $k$-th entry as $\widetilde{e}_k$ with torque $\widetilde{T}_{e_k}$, so that $\widetilde{T}_{e_1} \geq \widetilde{T}_{e_2} \geq \cdots \geq \widetilde{T}_{e_K}$. We then calculate the torque gap between two consecutive links by

$$G_{k,k+1} = \mu_k \times \frac{\widetilde{T}_{e_k}}{\widetilde{T}_{e_{k+1}} + \delta}, \tag{7}$$

where $\delta$ is a small constant to prevent division by zero, and the weight $\mu_k$ reflects the proportion of anomalous edges, those whose distance $D$, disparity $E$ and torque $T$ all exceed their respective means, that are captured within the top $k$ torque-ranked set, emphasizing the boundary between desirable and undesirable connections. The computation formula of $\mu_k$ is defined as

$$\mu_k = \frac{|High\_e \cap Top\_k|}{|High\_e|},$$
$$High\_e = \{e_k \doteq \langle i,j \rangle | D_{\langle i,j \rangle} \geq \bar{D}, E_{\langle i,j \rangle} \geq \bar{E}, T_{\langle i,j \rangle} \geq \bar{T}, \langle i,j \rangle \in \mathcal{E}\}, \tag{8}$$
$$Top\_k = \{e_k \doteq \langle i,j \rangle | Top_k\{T_{\langle i,j \rangle}\}, \langle i,j \rangle \in \mathcal{E}\},$$

where $\bar{D}, \bar{E}, \bar{T}$ denote the mean values of distance, disparity and torque, respectively, computed over all $K$ edges. The set $High\_e$ comprises edges exhibiting above-average values across all three metrics, while $Top\_k$ contains the top $k$ connections in TSL. According to Eq. 7, we can identify the optimal cutoff by locating the largest torque gap $\mathcal{K} = \arg\max_{1 \leq k \leq K-1} G_{k,k+1}$, which separates the edge set into two groups: undesirable connections $(\widetilde{e}_1, \cdots, \widetilde{e}_\mathcal{K})$ and desirable connections $(\widetilde{e}_{\mathcal{K}+1}, \cdots, \widetilde{e}_K)$.

In practice, multi-layer GNNs, such as APPNP Klicpera et al. (2019) and GCNII Chen et al. (2020), are widely adopted to enlarge the receptive field of graph convolutions. To enable each layer to adapt adjacency relationships based on evolving node features and capture different structural properties, we design a hierarchical rewiring strategy. Building on the torque formulation introduced above, we extend this mechanism across multiple propagation layers. In specific, for each layer $l$, we construct a dedicated propagation matrix that enables selective filtering of undesirable high-order interactions. Notably, to avoid misleading representations in the early stages of training, where unreliable representations could cause the model to discard informative neighbors or propagate spurious signals, rewiring at each layer is always performed with respect to the original input graph. Let $\mathbf{h}^{(l)}$ denote the node representation gained by the $l$-th layer. The torque is then recomputed as follows:

$$\mathbf{T}_{e_k}^{(l+1)} = (\mathbf{h}_i^{(l)} - \mathbf{h}_j^{(l)}) \times E_{\langle i,j \rangle}\mathbf{h}_j^{(l)}, \tag{9}$$

where $l = 0, \cdots, L-1$. Consequently, we gain the $(l+1)$-th order torque $T^{(l+1)}$ and the corresponding gap $G^{(l+1)}$ using Eqs. 6-8, from which we derive a pruned propagation matrix $\mathcal{A}^{(l+1)^*}$ with $(K - \mathcal{K})$ non-zero elements.

---

$^2$This form follows directly from the vector identity $\|\mathbf{a} \times \mathbf{b}\|^2 = \|\mathbf{a}\|^2\|\mathbf{b}\|^2 - (\mathbf{a} \cdot \mathbf{b})^2$.

**Edge-addition High-order Rewiring.** In the previous steps, we remove undesirable neighbors by computing the torque of existing edges based on two key attributes. Extending this strategy, we also consider expanding the receptive field by adding edges that are initially absent but potentially beneficial for message passing. However, evaluating torque across all missing edges is computationally intractable. We construct a candidate set $\mathcal{T}$ by selecting, for each node, its top-$t$ most similar peers using cosine similarity. We then compute the $(l + 1)$-th order torque $T^{(l+1)}$ for the resulting $N \times t$ candidate edges, and select $r \times N \times t$ edges with the lowest torque values, where $r$ is a sampling ratio. Nevertheless, this hard selection process is inherently non-differentiable and thus cannot be used in gradient-based optimization. To overcome this, we adopt a Bernoulli-guided learnable sampling process. Specifically, the Gumbel-Softmax reparameterization trick Jang et al. (2017) is leveraged, which enables differentiable sampling by approximating discrete decisions with a continuous relaxation. For each candidate edge $k$, we define its logits $\boldsymbol{\pi}_k = [\pi_{k0}, \pi_{k1}]$, where $\pi_{k0} = T_{e_k}^{(l+1)}$ (discard) and $\pi_{k1} = 1 - T_{e_k}^{(l+1)}$ (select). Drawing independent noise $g_{kj} \sim$ Gumbel $(0, 1)$, the soft selection probabilities are computed via

$$p_{kj} = \frac{\exp\left(\frac{\log(\pi_{kj}) + g_{kj}}{\tau}\right)}{\sum_{m=0}^{1} \exp\left(\frac{\log(\pi_{km}) + g_{km}}{\tau}\right)}, \forall j = 0, 1, k \in \{1, 2, \ldots, N \times t\},$$

(10)

where $\tau$ is a temperature parameter controlling the sharpness of the Gumbel-Softmax distribution. $p_{k1}$ serves as a differentiable weight indicating the likelihood of selecting candidate edge $k$. Finally, we construct the rewired propagation matrix $\mathcal{A}^{(l+1)}$ by augmenting $\mathcal{A}^{(l+1)^*}$ with these probabilistically weighted candidate edges, followed by the standard renormalization procedure.

**Messaging Passing on Rewired Graph.** By rewiring the adjacency matrix $\mathbf{A}$ as described, each propagation layer is equipped with an expanded receptive field, allowing for the capture of more effective multi-level interactions. To evaluate the effectiveness of the proposed THR in capturing high-order information in multi-layer GNNs, we use the deep-based APPNP model as an example. Subsequent ablation studies and parameter analyses are conducted within this framework. Let $\mathcal{N}_i^{(l+1)} = \{v_j | \mathcal{A}_{\langle i,j \rangle}^{(l+1)} \neq 0\}$ denotes the refined $(l + 1)$-layer neighborhood of node $v_i$; then the forward propagation at the $(l + 1)$-th layer of APPNP can be reformulated as:

$$\mathbf{h}_i^{(l+1)} = \text{ReLU}\Big( \sum_{j \in \{v_i\} \cup \mathcal{N}_i^{(l+1)}} \alpha \mathbf{h}_j^{(l)} + (1 - \alpha)\mathbf{h}_i^{(0)} \Big).$$

(11)

Here, $\alpha$ controls the trade-off between the hidden representation and the residual connection. The initial representation $\mathbf{h}_i^{(0)} = \mathbf{x}_i \boldsymbol{\Theta}$, where $\boldsymbol{\Theta} \in \mathbb{R}^{d \times m}$, is computed through a linear transformation of the input feature $\mathbf{x}_i$. The final node representations from the last layer are passed through a fully connected layer parameterized by $\boldsymbol{\Phi} \in \mathbb{R}^{m \times c}$, yielding the predicted class probabilities. These predictions are compared against the ground-truth labels using a cross-entropy loss, which is minimized through gradient-based optimization.

## 5 COMPLEXITY ANALYSIS

The dominant computational cost of THR lies in: 1) Torque computation and graph rewiring. For each order $l$, we compute torque values only on the edges in $\mathcal{A}^{(l)}$, costing $\mathcal{O}(|\mathcal{A}^{(l)}|)$, and then sort these values in $\mathcal{O}(|\mathcal{A}^{(l)}| \log |\mathcal{A}^{(l)}|)$. When adding edges, if the candidate set size is $B$, the combined probability calculation and sorting cost is $\mathcal{O}(B + B \log B)$. 2) Message passing on the rewired graph $\mathcal{A}^{(l)}$. For the input layer with parameter $\boldsymbol{\Theta} \in \mathbb{R}^{d \times m}$ on $\mathbf{X} \in \mathbb{R}^{N \times d}$, it costs $\mathcal{O}(Ndm)$. Aggregation over $\mathcal{A}^{(l)}$ then costs $\mathcal{O}(m|\mathcal{A}^{(l)}|)$ per layer. The output layer with $\boldsymbol{\Phi} \in \mathbb{R}^{m \times c}$ requires $\mathcal{O}(Nmc)$. Putting these costs for an $L$-layer network and assuming $B \ll |\mathcal{A}^{(l)}|$ for all $l$, the overall complexity is $\mathcal{O}(Ndm + \sum_{l=1}^{L} |\mathcal{A}^{(l)}| \log |\mathcal{A}^{(l)}|)$, which is slightly higher than that of standard methods with $\mathcal{O}(Ndm + m \sum_{l=1}^{L} |\mathcal{A}^{(l)}|)$.

## 6 EXPERIMENTS

**Datasets.** We evaluate our method on eleven standard node classification benchmarks, which include six heterophilous datasets: Texas, Wisconsin, Cornell, Actor, Penn94 and Flickr; five homophilous graphs: Citeseer, Cora, Pubmed, Tolokers and Questions. Among them, Tolokers, Questions, Penn94 and Flickr are large-scale datasets. The statistics for these datasets are summarized in Table 1, with further details provided in Appendix B.2.

**Baselines.** THR is a plug-in module that can be integrated into various state-of-the-art GNNs. To evaluate the improvements offered by THR for GNNs, we conduct experiments on three representative models: two models designed for homophilous graphs, namely the vanilla GCN Kipf & Welling (2017) and the deep-based APPNP Klicpera et al. (2019), as well as GPRGNN Chien et al. (2020), which is designed for heterophilous graphs.

Table 1: Benchmark dataset statistics.

| Datasets | Node Hom. | #Nodes | #Edges | #Classes | #Features |
|----------|-----------|--------|--------|----------|-----------|
| Texas | 0.11 | 183 | 295 | 5 | 1,703 |
| Wisconsin | 0.21 | 251 | 466 | 5 | 1,703 |
| Cornell | 0.30 | 183 | 280 | 5 | 1,703 |
| Actor | 0.22 | 7,600 | 26,752 | 5 | 931 |
| Citeseer | 0.74 | 3,327 | 4,676 | 7 | 3,703 |
| Cora | 0.81 | 2,708 | 5,278 | 6 | 1,433 |
| Pubmed | 0.80 | 19,717 | 44,327 | 3 | 500 |
| Tolokers | 0.60 | 11,758 | 51,900 | 2 | 10 |
| Questions | 0.84 | 48,921 | 153,540 | 2 | 301 |
| Penn94 | 0.47 | 41,554 | 1,362,229 | 2 | 4,814 |
| Flickr | 0.32 | 89,250 | 2,724,458 | 7 | 500 |

To evaluate the effectiveness of THR in comparison to other graph rewiring techniques, we select five superior methods, including First-order Spectral Rewiring (FoSR) Karhadkar et al. (2022), Batch Ollivier-Ricci Flow (BORF) Nguyen et al. (2023), Stochastic Jost and Liu Curvature Rewiring (SJLR) Giraldo et al. (2023), Deep Heterophily Graph Rewiring (DHGR) Bi et al. (2024) and randomly edge removal (DropEdge). Here, we adopt layer-wise DropEdge (Dropedge-L), as proposed by Rong et al. (2019), to ensure a fair comparison with the hierarchical structure of THR. Further details on all methods are provided in Appendix B.3.

**Setups.** We report node classification accuracy (ACC), defined as the proportion of correctly predicted labels. For all benchmark datasets, models are trained using the Adam optimizer. *Competitors are performed based on their respective source code.* Detailed hyperparameters and environment configurations for THR are provided in Appendix B.5, including the code link in B.1. Following prior work, we adopt the following data split strategy for all methods: 48% of the nodes are used for training, 32% for validation, and the remaining 20% for testing. Each experiment is conducted over 10 runs with different random splits, and the results are reported as the mean and standard deviation.

Table 2: Node classification results on benchmark datasets with GCN and GPRGNN as the backbone models: Mean ACC % (Standard Deviation %). The first- and second-best results are highlighted in **red** and **green**, respectively.

| Methods/Datasets | Citeseer | Cora | Pubmed | Texas | Wisconsin | Actor | Cornell |
|------------------|----------|------|--------|-------|-----------|-------|---------|
| GCN | 75.52 (2.19) | 86.96 (1.27) | 86.43 (0.38) | 58.61 (7.18) | 52.60 (8.72) | 30.15 (1.03) | 57.50 (4.66) |
| FoSR | 78.03 (1.45) | **87.00 (1.21)** | 86.34 (0.31) | **74.70 (6.23)** | 65.58 (4.89) | 30.16 (1.03) | 54.59 (5.01) |
| BROF | 78.45 (1.52) | 86.86 (1.35) | 86.42 (0.38) | 74.51 (6.26) | 65.59 (4.52) | 30.20 (1.17) | **60.27 (3.64)** |
| SJLR | 77.87 (1.81) | 86.60 (1.64) | **86.52 (1.73)** | 60.14 (0.89) | 55.16 (0.95) | 30.80 (1.34) | 58.11 (6.86) |
| DHGR | **78.68 (1.51)** | 86.61 (1.73) | 86.40 (0.38) | 60.20 (6.39) | **66.07 (12.51)** | **34.39 (0.99)** | 58.68 (5.01) |
| DropEdge-L | 74.93 (1.85) | 86.62 (1.23) | 83.07 (2.58) | 62.74 (8.32) | 58.82 (8.24) | 32.97 (0.92) | 54.32 (3.72) |
| THR | **80.43 (1.52)** | 86.97 (1.19) | **87.21 (0.45)** | 76.27 (4.67) | 68.09 (2.71) | **33.20 (0.90)** | 58.91 (9.11) |
| GPRGNN | 77.37 (1.83) | 87.34 (1.14) | 87.21 (0.43) | 89.22 (5.56) | 87.94 (5.29) | 37.27 (1.16) | 80.27 (6.63) |
| FoSR | 77.37 (1.83) | **87.52 (1.63)** | 87.22 (0.46) | 90.20 (5.04) | **89.85 (3.45)** | 37.25 (1.19) | 84.05 (7.88) |
| BORF | **78.77 (1.67)** | 87.49 (1.24) | 87.17 (0.39) | **91.16 (5.15)** | 89.11 (4.32) | 37.52 (1.06) | **85.49 (4.83)** |
| SJLR | 78.38 (1.49) | 86.97 (1.63) | **88.11 (0.41)** | 90.00 (2.83) | 89.26 (6.38) | 34.87 (1.69) | 81.62 (9.35) |
| DHGR | 77.77 (2.06) | 87.19 (1.39) | 87.69 (0.47) | 89.02 (4.31) | 86.03 (6.32) | 35.20 (1.20) | 84.31 (4.56) |
| DropEdge-L | 78.73 (1.91) | 86.91 (1.07) | 87.50 (0.48) | 90.17 (3.06) | 87.79 (6.28) | **37.77 (1.16)** | 84.05 (9.00) |
| THR | **79.15 (1.69)** | **87.60 (1.15)** | **88.28 (0.52)** | **91.96 (3.76)** | **91.91 (4.75)** | **38.00 (0.56)** | **86.22 (5.19)** |

**Node Classification Results.** Table 2 presents the test-set accuracy gains achieved by various rewiring approaches on GCN and GPRGNN across seven benchmark datasets. The comparison results for APPNP are provided in Appendix B.4. Several key insights can be drawn: 1) Compared to the baselines, all rewiring methods show performance improvements on most datasets, with particularly

notable gains on heterophilous graphs. 2) In all datasets, the proposed THR ranks among the top two performers, achieving the highest accuracy gain on the majority of benchmarks. 3) Although FoSR, BORF, and DHGR also exhibit strong performance on certain datasets, their gains are only marginally higher than those of THR. Overall, THR outperforms these methods and delivers the best results in all cases when GPRGNN is used as the downstream model. 4) DropEdge-L, which is also based on hierarchical graph rewiring, outperforms other rewiring methods on some datasets (e.g., Texas and Actor), validating the effectiveness of the hierarchical strategy. Although DropEdge-L shows performance improvements on certain datasets, its inherent randomness negatively impacts the model's performance, resulting in lower performance than the baseline in some cases, e.g., Citeseer. This further validates the effectiveness of the proposed torque-driven hierarchical approach.

Table 3: Node classification results on **large-scale** datasets: Mean ACC % (ROC AUC for imbalanced Questions and Tolokers) (Standard Deviation %), where the optimal and suboptimal results are highlighted in **red** and **green**, respectively. OoM means that the model suffers from the out-of-memory error.

| Methods/Datasets | Questions | Tolokers | Penn94 | Flickr |
|---|---|---|---|---|
| GCN | **75.26 (0.84)** | 83.79 (0.74) | 80.18 (0.36) | 63.52 (3.82) |
| FoSR | 75.19 (0.71) | **84.14 (0.99)** | 80.19 (0.35) | 63.74 (4.11) |
| BROF | 75.15 (0.84) | MemoryError | OoM | OoM |
| SJLR | 72.07 (6.12) | 84.14 (1.14) | **80.20 (0.28)** | **64.49 (2.82)** |
| DHGR | OoM | 83.45 (2.16) | OoM | OoM |
| DropEdge-L | 74.06 (1.11) | 84.00 (0.65) | 62.27 (0.35) | 63.10 (3.22) |
| THR | **75.92 (1.09)** | **84.43 (0.88)** | **80.32 (0.23)** | **67.53 (2.03)** |
| GPRGNN | 72.89 (1.42) | 71.99 (0.93) | 84.18 (0.30) | 61.05 (4.24) |
| FoSR | 72.91 (1.43) | **71.99 (0.93)** | **84.22 (0.29)** | **62.35 (3.62)** |
| BORF | **72.99 (1.44)** | MemoryError | OoM | OoM |
| SJLR | 72.27 (1.24) | 69.46 (1.07) | 83.89 (0.20) | 61.61 (3.22) |
| DHGR | OoM | 70.96 (1.14) | OoM | OoM |
| DropEdge-L | 72.07 (1.35) | 71.98 (1.09) | 83.67 (0.44) | 60.77 (3.53) |
| THR | **73.41 (0.98)** | **72.05 (1.24)** | **84.45 (0.29)** | **64.18 (1.37)** |

**Results on Larger Graphs.** Scalability of rewiring techniques on large graphs is crucial, particularly for end-to-end methods that dynamically add and remove edges during training. In THR, the primary computational cost arises from computing torques and the corresponding gaps, which incurs a complexity of $\mathcal{O}(|\mathcal{A}^{(l)}|\log|\mathcal{A}^{(l)}|)$ (see Section 5 for details). Despite this overhead, THR remains computationally feasible for large graphs. Table 3 compares several rewiring schemes on larger datasets, with THR consistently outperforming all alternatives. Notably, with the exception of the Flickr dataset, all rewiring methods show only marginal improvements, and in some cases, even lead to a decline in performance. This can be attributed to the fact that the raw graphs of these datasets already contain sufficient structural information, and the rewiring methods introduce only minor modifications. Moreover, in some instances, they result in the loss of critical semantics, which negatively impacts classification performance.

**Ablation Study.** We conduct an ablation study to assess the impact of edge removal and addition operations in THR, using GCN, GPRGNN, and APPNP as backbone models. THR has three variants: edge-addition THR (A-THR), edge-removal THR (R-THR), and mixed THR (M-THR). As illustrated in Figure 2, most rewiring variants significantly outperform their base GNNs. However, on the Wisconsin dataset, GPRGNN slightly surpasses GPRGNN with R-THR, likely because GPRGNN effectively allocates signed edges to distinguish class information, while R-THR removes heterophilous links, inadvertently causing the model to lose some discriminative features. On the Flickr dataset, R-THR improves per-

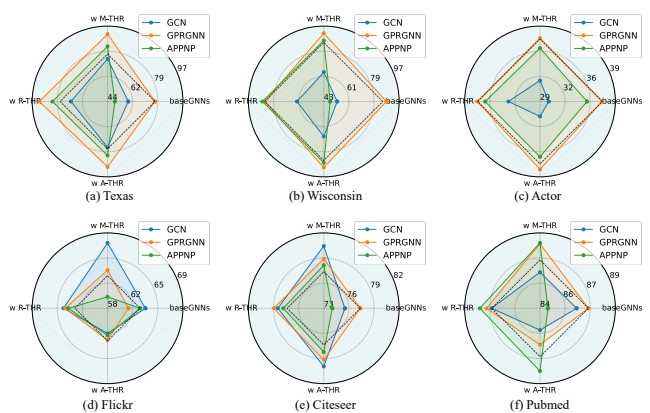

Figure 2: Ablation study: Performance comparison of GCN, GPRGNN, and APPNP with various THR variants across six datasets.

formance for all models, while A-THR and M-THR degrade the performance of APPNP. Similarly, on PubMed, both A-THR and M-THR reduce the performance of GCN. These results suggest that, for these graphs, excessive edge addition leads to information interference and confusion of node features, whereas GPRGNN mitigates this effect through its signed edge strategy. In conclusion,

the THR strategy excels in improving model performance, although it adopted to achieve optimal performance varies across datasets with different characteristics.

Moreover, to investigate the significance of the proposed torque, which integrates feature distance and homophily ratio disparity from a physical perspective, we evaluate THR and its variants based on the edge-removal strategy. $THR_{dis.}$ refers to the rewiring method based solely on the distance metric between node pairs. $THR_{torque\ w/o\ homo.}$ leverages the torque without considering the disparity in homophily ratio. $THR_{w/o\ H}$ denotes the version of THR without hierarchical rewiring, where all layers share the same graph. Table 4 displays the ablation results, showing that the node classification accuracy of variants that do not use the proposed torque decreases across all heterophilous datasets. Moreover, on the Cornell and Flickr datasets, $THR_{w/o\ H}$ outperforms THR, suggesting that layer-wise rewiring may excessively complicate their graph structures, thereby hindering the propagation of effective information. In summary, both THR and $THR_{w/o\ H}$ rely on the proposed torque for graph rewiring and both rank in the top two across all datasets. This validates that THR effectively models heterophilous graphs by integrating the distance and homophily ratio disparities between node pairs from a physical perspective.

Table 4: Ablation study: A comparison of THR and its variants by removing specific components. The optimal and suboptimal results are highlighted in bold and underlined, respectively.

| Datasets | Texas | Wisconsin | Cornell | Actor | Penn94 | Flickr |
|---|---|---|---|---|---|---|
| $THR_{dis.}$ | 70.39 (9.62) | 74.56 (7.21) | 76.22 (8.53) | 35.80 (1.27) | 74.98 (0.55) | 61.51 (4.32) |
| $THR_{torque\ w/o\ homo.}$ | 67.84 (10.96) | 72.94 (8.71) | 76.03 (8.40) | 35.57 (1.31) | 75.94 (0.57) | 61.23 (4.74) |
| $THR_{w/o\ H}$ | 70.98 (8.12) | 75.29 (4.45) | **78.65 (6.78)** | 35.96 (1.33) | 76.14 (0.63) | **64.13 (2.27)** |
| THR | **72.01 (6.13)** | **75.89 (3.46)** | 77.30 (7.17) | **36.28 (1.15)** | **76.21 (0.47)** | 63.28 (1.56) |

**Parameter Analysis.** Since the edge-removal procedure automatically determines the cutoff $\mathcal{K}$, we investigate the main hyperparameter $t$ of THR, which defines the number of candidate edges for addition. As shown in Figure 3, we present the performance curves for varying $t$ values in $\{2, 4, 6, 8, 10\}$ across five datasets. On both homophilous and heterophilous datasets, accuracy increases as $t$ grows, demonstrating that the proposed edge-addition scheme aids the model in capturing global information. However, this does not imply that adding more edges is always beneficial. For instance, on the Flickr dataset, performance decreases when $t = 8$, as excessive edge addition may introduce noise, as highlighted in the ablation study. Sensitivity analysis of $\alpha$ and $L$ is provided in Appendix B.4.

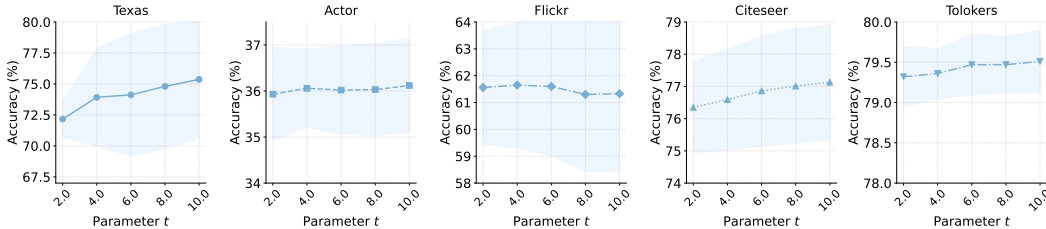

Figure 3: Parameter sensitivity: Performance curves on five datasets as the number of candidate edges $t$ varies from 2 to 10.

## 7 CONCLUSION

In this paper, we proposed a **Torque**-driven **H**ierarchical **R**ewiring strategy (THR), which dynamically refined the graph structures to enhance representation learning on heterophilous and homophilous graphs. By introducing an interference-aware torque metric, the product of the displacement vector and the feature vector weighted by the homophily ratio disparity, THR automatically removed undesirable connections and introduced beneficial ones during message passing. This hierarchical rewiring yielded interference-resilient, importance-aware propagation tailored to each layer's receptive field. Extensive evaluations across homophilous and heterophilous benchmark datasets demonstrated that THR consistently obtained the performance gains and outperformed other rewiring methods.

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

## A  ALGORITHM

Algorithm 1 outlines the complete workflow of APPNP with THR.

---
**Algorithm 1:** GNN with THR

---
**Input:** Node features $\{\mathbf{x}_i \in \mathbb{R}^d\}_{i=1}^N$, candidate edge set $\mathcal{T}$, ground truth matrix $\mathbf{Y}$, the number of layers $L$, hyperparameters $t$ and $\alpha$.

**Output:** The predicted class label.

1   Initialize network parameters $\mathbf{\Theta}, \mathbf{\Phi}$;

2   $\mathbf{h}_i^{(0)} = \text{ReLU}(\mathbf{x}_i \mathbf{\Theta})$;

3   **while** *not convergent* **do**

4     **for** $l = 1 \rightarrow L$ **do**

5       $\triangleright$ **Forward Propagation**

6       Compute pairwise distance $D_{\langle i,j \rangle}^{(l)}$ and homophily ratio disparity $E_{\langle i,j \rangle}$ with Eqs. 4 and 5;

7       Compute the $l$th order torques with Eq. 6 and sort them. // Torque computation

8       Gain the largest torque gap $\mathcal{K}$ with Eqs. 7-8;

9       Remove the top $\mathcal{K}$ edges to gain $\mathcal{A}^{(l)^*}$. // Removing undesirable edges

10       Compute the sampling probability of candidate edges with Eq. 10;

11       Add beneficial candidate edges to form the refined propagation matrix $\mathcal{A}^{(l)}$. // Adding desirable connections

12       Update node representation $\mathbf{h}_i^{(l)}$ with Eq. 11. // Message passing

13       $\triangleright$ **Backward Propagation**

14       Classifier $f(\cdot) \longleftarrow \text{LocalUpdating}(\mathbf{x}_i, \{\mathcal{A}^{(l)}\}_{l=1}^L)$ with the cross-entropy loss // Standard training

15     Obtain $\hat{\mathbf{y}}_i = \text{Softmax}\left(\mathbf{h}_i^{(L)} \mathbf{\Phi}\right)$;

16   **return** The predicted class label of the $i$-th node is given by $\arg\max \hat{\mathbf{y}}_i$.

---

## B   MORE EXPERIMENTAL RESULTS

### B.1   CONFIGURES

We construct a series of experiments to assess the proposed THR. Our model is implemented in PyTorch on a workstation with AMD Ryzen 9 5900X CPU (3.70GHz), 64GB RAM and RTX 3090GPU (24GB caches). Our code is available at `https://anonymous.4open.science/r/THR-FE0B/README.md`.

### B.2   DATASETS

- Homophilous Datasets. Citeseer, Cora and Pubmed are three citation networks, and they are published in Sen et al. (2008). Specifically,

  - **Citeseer** comprises 3,327 publications classified into six categories, with each paper encoded by a 3,703-dimensional binary word-presence vector.
  - **Cora** consists of 2,708 scientific publications classified into seven research topics. Each paper is represented by a 1,433-dimensional binary feature vector indicating the presence of specific word.
  - **Pubmed** is a larger citation network of 19,717 diabetes-related articles labeled among three classes. Papers are described by 500-dimensional term frequency–inverse document frequency feature vectors, and citation edges capture scholarly references.
  - **Tolokers** Platonov et al. (2023) is built from the Toloka crowdsourcing platform, comprising 11,758 nodes and 519,000 edges that link workers who collaborated on the same task. Each node carries a 10-dimensional feature vector and is assigned one of two labels based on whether the worker was banned.
  - **Questions** Platonov et al. (2023) is an interaction graph of users on the Yandex Q question-answering platform, comprising 48,921 nodes and 153,540 edges that link users who interacted on the same question. Each node carries a 301-dimensional feature vector and a binary label for node classification.

- Heterophilous Datasets

  - **Texas**, **Wisconsin**, **Cornell** are WebKB datasets used in Pei et al. (2020), where nodes correspond to individual web pages and edges correspond to the hyperlinks between them. Every node is described by a bag-of-words feature vector extracted from its page content, and each page has been manually labeled into one of five categories.
  - **Actor** Tang et al. (2009) is the actor-only induced subgraph of a film–director–actor–writer network on Wikipedia, where each node represents an actor and an undirected edge connects two actors if they co-occur on the same Wikipedia page.
  - **Penn94** Lim et al. (2021) is a subgraph of the Facebook100 dataset featuring 41,554 university students as nodes, connected by 1,362,229 undirected friendship edges. Each node is described by a five-dimensional feature vector and labeled by the gender of the students.
  - **Flickr** Zeng et al. (2020) is an undirected graph originated from NUS-wide, including 89,250 nodes and 2,724,458 edges. Each node is an image with 500-dimensional bag-of-word features and each edge links two images sharing some common properties.

### B.3   BASELINES

#### B.3.1   GNNS FOR HOMOPHILOUS AND HETEROPHILOUS GRAPHS

**GCN** generalize convolutional neural networks to graph-structured data by iteratively aggregating feature information from each node's local neighborhood,

$$\mathbf{h}_i^{(l)} = \sigma(\widetilde{\mathbf{A}}\mathbf{h}_i^{(l-1)}\mathbf{W}^{(l)}), \qquad (12)$$

where $\mathbf{W}$ is the learnable parameter matrix.

**APPNP** first achieves the feature transformation by:

$$\mathbf{H}^{(0)} = \mathbf{XW}, \tag{13}$$

and then propagating message via a Personalized PageRank scheme:

$$\mathbf{H}^{(l)} = (1 - \alpha)\mathbf{PH}^{(l-1)} + \alpha\mathbf{H}^{(0)}. \tag{14}$$

Here, $\mathbf{P} = \mathbf{D}^{-1/2}\mathbf{A}\mathbf{D}^{-1/2}$ is the symmetrically normalized adjacency matrix and $\alpha$ is a trade-off hyperparameter.

**GPRGNN** generalizes personalized PageRank by treating each hop's contribution as a learnable parameter:

$$\mathbf{H} = \sum_{l=1}^{L} \gamma^l \mathbf{PH}^{(0)}, \mathbf{H}^{(0)} = \mathbf{HW}, \tag{15}$$

where $\gamma^l \mathbf{P}$ measures the propagation coefficient for the connection between nodes $v_i$ and $v_j$.

### B.3.2 REWIRING STRATEGIES

**DropEdge** randomly removes edges at each training epoch to act as both data augmentation and message-passing reduction, which is used to mitigate over-fitting and over-smoothing problems.

**FoSR** is a preprocessing method, which aims to address the over-squashing issue by improving the graph connectivity. It adds edges by exploring the first order change in the spectral gap.

**BORF** uses the Ollivier-Ricci curvature to rewire graph, where minimally curved edges causing the information bottlenecks should add connections and maximally curved edges leading to over-smoothing should be removed.

**SJLR** combines the Jost–Liu Curvature of each edge with the embedding similarity between its incident nodes, and uses the weighted score as the probability for edge removal or addition.

**DHGR** compares the neighborhood feature distribution and neighborhood label distribution between node pairs; edges connecting nodes with low similarity (heterophilous) are pruned, while edges between highly similar (homophilous) nodes are added.

**DHGR vs. THR**. Although both methods essentially assess edge homophily or heterophily through feature and label differences, they follow distinct methodological lines. DHGR is a preprocessing approach that aggregates neighborhood features and derives local label distributions from pseudo-labels produced by a pre-trained model, a heuristic design without explicit theoretical grounding. By contrast, THR operates within the optimizing model, contrasting node representations and quantifying their homophily ratio disparity, thereby aligning with prior theoretical proofs and offering a more principled formulation.

### B.4 EXPERIMENTS

**Classification Results.** Table 5 shows the performance gains brought by APPNP with diverse rewiring methods. We can observe that on most datasets, THR obtains the optimal performance, indicating its effectiveness.

**Parameter Sensitivity.** Although $\alpha$ balancing the contribution of the learned high-order representation and the original input features originates from APPNP, THR modifies the graph structure over which propagation occurs. To examine how signal diffusion changes with respect to $\alpha$ under the rewired graph, we perform a sensitivity analysis shown in Figure 4, where a larger $\alpha$ increases the influence of the hidden representations. We observe that, for smaller heterophilous graphs (e.g., Texas and Actor), optimal accuracy is achieved at $\alpha = 0.05$, implying that raw node features provide sufficient discriminative power. In contrast, on larger or homophilous graphs, better performance is observed when $\alpha = 0.5$, reflecting the necessity of hidden representations to capture more complex community structures. Moreover, for all datasets, the best results are gained at a larger $\alpha$, which demonstrates the effectiveness of excavating deep features.

Figure 5 explores the effect of network depth $L$. For small graphs (Texas, Citeseer and Film), performance improves as the number of layers increases, since deeper networks are required to

Table 5: Node classification results on benchmark datasets with APPNP as the backbone model: Mean ACC % (Standard Deviation %). The first- and second-best accuracies are highlighted in bold and underlined, respectively.

| Methods/Datasets | APPNP | FoSR | BROF | SJLR | DHGR | DropEdge-L | THR |
|---|---|---|---|---|---|---|---|
| Texas | 49.17±3.30 | 78.04 (3.70) | 73.53 (7.66) | 81.57 (3.94) | 78.04 (4.62) | 72.75 (4.68) | **84.12 (3.22)** |
| Wisconsin | 47.60±4.54 | 74.26 (4.47) | 75.00 (4.41) | 83.53 (5.95) | 73.82 (3.77) | 71.91 (4.85) | **87.79 (3.54)** |
| Actor | 35.24 (0.56) | 35.51 (1.42) | 35.35 (1.28) | 35.19 (1.13) | 35.90 (1.16) | 35.48 (1.12) | **36.34 (0.87)** |
| Cornell | 67.57 (5.54) | 67.57 (5.54) | 68.38 (7.46) | 74.59 (5.16) | 69.46 (8.80) | 69.19 (7.27) | **77.57 (8.02)** |
| Penn94 | 76.53 (0.28) | 76.53 (0.28) | OoM | 79.73 (0.24) | 79.10 (0.40) | 76.13 (0.40) | **82.56 (0.43)** |
| Flickr | 57.26 (7.69) | 56.31 (6.99) | OoM | 61.86 (5.17) | 62.20 (1.10) | 61.25 (3.30) | **63.28 (1.56)** |
| Citeseer | 74.02 (0.38) | 77.65 (1.55) | 77.65 (1.24) | 77.25 (1.35) | 76.89 (1.81) | 77.69 (1.67) | **78.74 (1.29)** |
| Cora | 85.89 (1.19) | 85.89 (1.19) | 85.19 (1.87) | **86.51 (1.59)** | 85.85 (1.79) | 85.54 (1.14) | 86.35 (1.61) |
| Pubmed | 87.19 (0.55) | 87.19 (0.55) | 87.16 (0.40) | **88.84 (0.40)** | 88.47 (0.44) | 87.66 (0.33) | 88.31 (0.49) |
| Tolokers | 75.11 (0.74) | 75.11 (0.74) | OoM | 78.46 (1.11) | 75.33 (0.83) | 74.64 (1.06) | **79.29 (0.42)** |

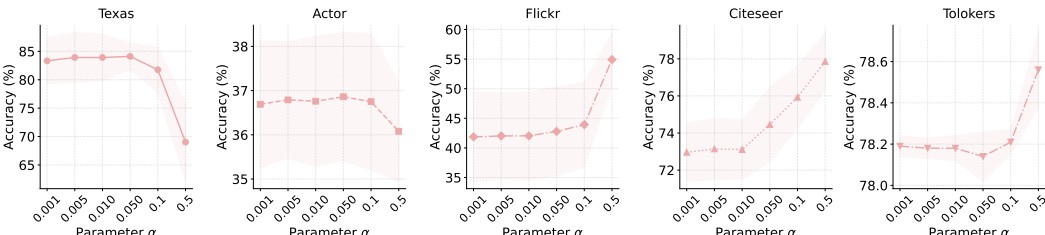

Figure 4: Parameter sensitivity: Performance curves on five datasets with layers changing in $\{2, 4, 8, 16, 32\}$.

capture sufficient high-order information. In contrast, for large graphs (Tolokers and Flickr), the best performance is achieved with only two layers, indicating that shallow message passing already provides sufficiently discriminative representations. However, while APPNP can alleviate over-smoothing to some extent, it does not explicitly address this issue on these graphs; overcoming depth-related bottlenecks therefore remains an open direction for future research.

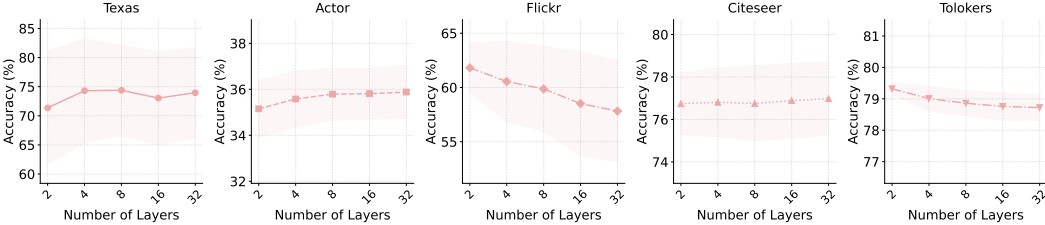

Figure 5: Parameter sensitivity: Performance curves on five datasets with layers changing in $\{2, 4, 8, 16, 32\}$.

### B.5 HYPERPARAMETERS

In this subsection, we present the detailed hyperparameters used in the experiments, which are also available in the code. The hyperparameters are summarized in Tables 7 to 8. "Lr" refers to the learning rate, "Wd" denotes the weight decay, and "PPR" is a specific hyperparameter used in GPRGNN.

## C  BROADER IMPACT STATEMENT

This study aims to enhance message passing in graph neural networks through graph rewiring. As a result, it contributes to better performance and broader applicability of GNNs across a wide range

Table 6: Hyperparameters of THR on GCN across 11 datasets.

| Datasets | Lr | Wd | Dropout | $L$ | $t$ | Normalize Data | Hidden Size |
|---|---|---|---|---|---|---|---|
| Texas | 0.05 | 0.0005 | 0.5 | 2 | 5 | Yes | 32 |
| Wisconsin | 0.05 | 0.0005 | 0.5 | 2 | 5 | Yes | 32 |
| Cornell | 0.05 | 0.0005 | 0.5 | 2 | 10 | Yes | 512 |
| Actor | 0.01 | 0.0005 | 0.5 | 2 | 2 | No | 32 |
| Citeseer | 0.01 | 0.0005 | 0.5 | 2 | 10 | Yes | 32 |
| Cora | 0.01 | 0.005 | 0.5 | 2 | 2 | No | 32 |
| Pubmed | 0.01 | 0.0005 | 0.5 | 2 | 2 | Yes | 32 |
| Tolokers | 0.005 | 0.0 | 0.2 | 2 | 1 | No | 32 |
| Questions | 0.005 | 0.0 | 0.2 | 5 | 1 | No | 32 |
| Penn94 | 0.001 | 5e-8 | 0.5 | 2 | 1 | No | 32 |
| Flickr | 0.01 | 0.0005 | 0.3 | 2 | 1 | Yes | 32 |

Table 7: Hyperparameters of THR on GPRGNN across 11 datasets.

| Datasets | Lr | Wd | Dropout | $L$ | $t$ | Normalize Data | PPR | Hidden Size |
|---|---|---|---|---|---|---|---|---|
| Texas | 0.05 | 0.0005 | 0.5 | 2 | 5 | Yes | 1 | 32 |
| Wisconsin | 0.05 | 0.0005 | 0.0 | 2 | 5 | Yes | 1 | 64 |
| Cornell | 0.05 | 0.0005 | 0.5 | 2 | 10 | Yes | 0.9 | 512 |
| Actor | 0.01 | 0.0 | 0.5 | 2 | 2 | Yes | 0.9 | 32 |
| Citeseer | 0.01 | 0.0 | 0.5 | 2 | 10 | Yes | 0.1 | 64 |
| Cora | 0.01 | 0.005 | 0.5 | 2 | 2 | Yes | 0.1 | 32 |
| Pubmed | 0.05 | 0.0005 | 0.5 | 2 | 2 | Yes | 0.2 | 32 |
| Tolokers | 0.005 | 0.0 | 0.5 | 2 | 1 | No | 0.1 | 256 |
| Questions | 0.05 | 5e-8 | 0.5 | 2 | 1 | No | 0.1 | 32 |
| Penn94 | 0.01 | 0.0001 | 0.5 | 2 | 1 | No | 0.9 | 32 |
| Flickr | 0.05 | 0.0005 | 0.5 | 2 | 1 | No | 0.9 | 32 |

Table 8: Hyperparameters of THR on APPNP across 11 datasets.

| Datasets | Lr | Wd | Dropout | $\alpha$ | $L$ | $t$ | Normalize Data | Hidden Size |
|---|---|---|---|---|---|---|---|---|
| Texas | 0.001 | 0.0005 | 0.7 | 0.05 | 8 | 5 | No | 512 |
| Wisconsin | 0.001 | 0.5 | 0.5 | 0.05 | 4 | 5 | No | 512 |
| Cornell | 0.001 | 0.05 | 0.7 | 0.5 | 8 | 2 | No | 512 |
| Actor | 0.001 | 0.05 | 0.1 | 0.5 | 8 | 5 | No | 512 |
| Citeseer | 0.001 | 0.05 | 0.4 | 0.5 | 8 | 10 | No | 512 |
| Cora | 0.001 | 0.5 | 0.4 | 0.5 | 4 | 2 | No | 512 |
| Pubmed | 0.01 | 5e-8 | 0.4 | 0.5 | 4 | 2 | No | 512 |
| Tolokers | 0.001 | 5e-8 | 0.1 | 0.8 | 2 | 2 | No | 512 |
| Questions | 0.001 | 5e-8 | 0.1 | 0.8 | 2 | 2 | No | 512 |
| Penn94 | 0.001 | 5e-8 | 0.1 | 0.8 | 2 | 2 | No | 512 |
| Flickr | 0.001 | 0.5 | 0.1 | 0.8 | 2 | 2 | No | 512 |

of tasks, including recommendation systems, molecular property prediction, traffic forecasting, and social network.

## D  THE USE OF LARGE LANGUAGE MODELS (LLMS)

In this paper, we use LLMs as a general-purpose assistive tool to polish writing. LLMs play a significant role in enhancing the clarity and overall quality of the paper. However, LLMs are not involved in research ideation and experimental design, and its contribution is limited to writing optimization and logical structure improvement.

