# OpenReview forum: "GraphTorque: Torque-Driven Rewiring Graph Neural Network"
_ICLR.cc/2026/Conference — ICLR 2026 Conference Withdrawn Submission_

### Official Review · Reviewer_5f9Z · 2025-10-24

**Soundness:** 2
**Presentation:** 2
**Contribution:** 2
**Rating:** 2
**Confidence:** 4

**Summary:**

The authors propose a graph rewiring strategy, inspired by mechanical torque, to improve message passing in GNNs under both homophily and heterophily. During training, each edge is assigned a score that guides rewiring, with Gumbel–Softmax parameterizing the selection, yielding a dynamic graph-rewiring method.

**Strengths:**

* The rewiring method combines (i) the distance between node representations and (ii) the neighbor’s feature strength, weighted by the local homophily gap a principled and interesting criterion.

* Unlike curvature or spectral based rewiring, the scoring is directly dependent on the task and the data, leveraging node features together with homophily information.


* The model’s computational complexity appears operational on large graphs, making it practical beyond small benchmarks.

**Weaknesses:**

- **Ambiguity in the introduction on the heterophily definition.**
  The introduction conflates label heterophily  with feature dissimilarity. Phrases such as “addressing heterophily, where nodes with dissimilar labels or features  tend to be connected” and “heterophilous graphs typically connect node pairs with low similarity” assume that heterophilous edges usually join feature dissimilar nodes, whereas one can have similar features but different labels. This ambiguity harms the motivation of the work.

- **Reliance on pseudo-labels for local homophily.**
  The gap \(|h_i^{+} - h_j^{+}|\) is computed with pseudo-labels beyond annotated nodes. This can create a kind of self-reinforcing error : early mistakes in pseudo-labels may be treated as correct, shaping subsequent estimates and making those errors more likely to persist. The risk could be higher with imbalance class. Additionally the paper notes progressive refinement, but it does not quantify how sensitive the method is to pseudo-label.

- **Limited theory behind the “torque” intuition.**
  The torque  analogy remains empirical. There is no formal guarantee linking the torque score to reduced over-squashing, or proxy analyses (e.g., Jacobian/sensitivity bounds, changes in effective resistance, or diagnostics of topological bottlenecks). The paper should clarify :
  - (i) How torque values flag edges that exacerbate over-squashing ?
  - (ii) How you can analyse the final rewired structure ?
  - (iii) Why these structural rewiring should mitigate over-squashing ?

In my view, this is the paper’s most significant weakness

- **Missing discussion of feature-aware rewiring baselines.**
  While the method leverages node features, the authors omit recent approaches that also use features for rewiring [1–4].
  To clarify the paper’s evaluation and highlight the method’s performance, a focused discussion and comparison (not with all baselines, of course) is in my view necessary. Note that [4] also uses the Gumbel–Softmax to perform dynamic rewiring based on triangle substructures.

  - [1] Hugo Attali, Buscaldi, D., & Pernelle, N. Delaunay Graph: Addressing over-squashing and over-smoothing using Delaunay triangulation, ICML 2024.
  - [2] Jonas Linkerhägner, Cheng Shi, Ivan Dokmanić : Joint Graph Rewiring and Feature Denoising via Spectral Resonance, ICLR 2025.
  - [3] Celia Rubio-Madrigal, Adarsh Jamadandi, Rebekka Burkholz : GNNs Getting ComFy: Community and Feature Similarity Guided Rewiring, ICLR 2025
  - [4] Hugo Attali, Thomas Papastergiou, Nathalie Pernelle, Fragkiskos D. Malliaros : Dynamic Triangulation-Based Graph Rewiring for Graph Neural Networks, CIKM 2025.

- **Notation issues.**
  The symbol \(D\) is overloaded (node degree vs. displacement vector). Use distinct symbols to avoid confusion and improve the clarity.

**Questions:**

-  See Weakness please.

I would welcome the opportunity to continue this discussion and engage with these points during the rebuttal phase.

---

### Official Review · Reviewer_Xk2o · 2025-10-27

**Soundness:** 1
**Presentation:** 2
**Contribution:** 2
**Rating:** 2
**Confidence:** 4

**Summary:**

The paper introduces a method that that rewires the graph structure of GNNs during training to improve node classification, especially on heterophilous graphs where neighbors mostly have different labels. It defines a "torque score" for each edge using the (learned) node embedding similarity, and how different their local "label distributions" are (with pseudo-labels during training). High "torque" edges are considered harmful and get pruned, while low "torque" edges can be added; they only consider a candidate subset given by cosine similarity for computational reasons. The rewiring is done separately at every layer.

**Strengths:**

- The paper presents an integrated, train-time rewiring mechanism instead of doing static preprocessing.
- The method performs rewiring separately at each layer, which is an important design choice albeit not novel.
- It includes ablations that justify each component of the method.

**Weaknesses:**

1. The motivation for this rewiring method is lacking. In parts of the paper, "bad" edges are described as spurious or missing, but in the experiments the method is compared against approaches targeting over-squashing, which is a different problem.
2. The paper relies almost entirely on the outdated assumption that "heterophily = bad" and does not present any alternative conceptual motivation or new theoretical insight beyond that.
3. The claim that adversarial edges tend to connect low-similarity nodes does not imply the converse, that low-similarity edges are adversarial or spurious. Moreover, there are no ablations or experiments on robustness or resistance to adversarial edges, even though that is presented as a core motivation for the method.
4. Importing a concept from physics is not a contribution by itself, unless that concept enables new insight, which is not the case here.
5. It looks like many of the experimental gains fall within the standard deviations.
6. The paper does not discuss or compare to prior work on similarity-based rewiring methods (1), layer-wise rewiring methods (2) or noisy edge rewiring methods (3).
7. Minor: Fig.4 caption seems to be wrong, as it is the same one than Fig.5.

(1) GNNs Getting ComFy: Community and Feature Similarity Guided Rewiring. Celia Rubio-Madrigal et al. ICLR 2025.

(2) DRew: Dynamically Rewired Message Passing with Delay. Benjamin Gutteridge et al. ICML 2023.

(3) Towards Understanding and Reducing Graph Structural Noise for GNNs. Mingze Dong et al. ICML 2023.

**Questions:**

1. What problem is this method actually solving: over-squashing or noisy / spurious edges? If it is over-squashing, there is no theoretical link to their methodological proposal. If it is noisy edges, there is no experimental evidence in terms of robustness, and no comparison against prior methods with that stated goal.
2. How does the method affect model overconfidence, given that it reinforces pseudo-labels produced by the model itself?
3. Why are higher torque values assumed to indicate less reliable edges? This looks like a fallacy given by the "adversarial edge" story, which does not necessarily follow. Also, "reliability" is never formally defined, and there is no theory that explains this connection to the method.
4. How does this criterion compare to just maximizing pairwise similarity, which already exists, e.g. (1)? And how about comparing on equal footing to a layer-wise similarity-based rewiring strategy using the learned embeddings during training? Why is the torque view necessary?

---

### Official Review · Reviewer_moV6 · 2025-10-28

**Soundness:** 3
**Presentation:** 3
**Contribution:** 4
**Rating:** 4
**Confidence:** 4

**Summary:**

This work introduces Torque-driven Hierarchical Rewiring as a novel rewiring strategy to overcome the over-squashing problem in GNNs. THR assigns a torque value, a physics-inspired concept, to each edge. This metric measures the potential of each edge to interfere with message passing based on node feature differences and local label homophily. Using these torque values, the method hierarchically rewires the graph by pruning unreliable edges and sampling important connections in an end-to-end manner.

**Strengths:**

- This is the first work that uses a physics-inspired torque idea as a graph rewiring technique; this is novel and well motivated by the successful application of torque in other fields.
- Experiments show strong performance across different evaluation setups.
- Design choices are thoroughly ablated.

**Weaknesses:**

- The method’s main limitation lies in its relatively higher computational complexity, primarily due to the need to sort edges after torque computation in order to identify the torque gap for edge removal. Of course, this could not be a limiting factor in practice, but the authors do not provide a runtime comparison. I would highly recommend a runtime comparison be included in the Supplementary.
- The authors have mainly performed experiments on node-level dataset. However, a lot of the works on graph rewiring, over-smoothing, over-squashing, and long-range dependencies are also evaluating on graph-level datasets (such as the LRGB [[2], [3]]) or other synthetic tasks (such as Trees-NeighborsMatch [[4]]). Does the proposed method support these datasets? Could the authors provide a comparison on these datasets? I would also recommend that the authors use all of the datasets from [[5]] for heterophilic tasks.
- Did you conduct an ablation study comparing the torque-based edge removal to a simpler strategy that removes a fixed number of edges with the lowest torque values? It would be interesting to observe the performance trade-off in such a setting. If the performance degrades substantially, the higher computational complexity of the proposed approach would be justified.
- Have you evaluated the method’s behavior when excluding initial input graph features? Understanding how this affects training stability would help clarify the model’s dependence on early representations.
- Could the proposed approach be extended to dynamic graphs, where edges appear/disappear over time, as in DGraph [[1]]? The concept of an adaptive receptive field seems promising for this.
- (Minor) The method’s performance may appear to depend on reliable early-stage guidance, as noisy or unstable initial representations may negatively affect the rewiring process.
- (Minor) The font on Figure 2 is very small. Please consider increasing the font.
- (Very minor) Please consider using \citep{} when referencing a paper, but not directly referring to the authors.

[1]: https://arxiv.org/abs/2207.03579
[2]: https://arxiv.org/abs/2206.08164
[3]: https://arxiv.org/abs/2309.00367
[4]: https://arxiv.org/abs/2006.05205
[5]: https://arxiv.org/pdf/2302.11640

**Questions:**

Please see weaknesses above.

Overall, I believe that the work is interesting and that the paper could be accepted. However, I suggest that the authors add a runtime analysis, such that it would be easier to understand the performance tradeoffs. Moreover, a clarification regarding graph-level tasks and other empirical results on synthetic datasets would be needed --- I recommend that the authors add results on the LRGB [[2], [3]] and some synthetic setups (such as Trees-NeighborsMatch [[4]] for over-squashing). I would also highly recommend that the authors provide results on all of the datasets from [[5]].


[1]: https://arxiv.org/abs/2207.03579
[2]: https://arxiv.org/abs/2206.08164
[3]: https://arxiv.org/abs/2309.00367
[4]: https://arxiv.org/abs/2006.05205
[5]: https://arxiv.org/pdf/2302.11640

---

### Official Review · Reviewer_VcxD · 2025-10-30

**Soundness:** 2
**Presentation:** 2
**Contribution:** 2
**Rating:** 2
**Confidence:** 4

**Summary:**

This paper proposes an edge rewiring strategy that considers both the similarity of node features and the consistency of labels (or pseudo-labels). The authors evaluate the proposed rewiring method on both homophilic and heterophilic datasets, demonstrating its effectiveness compared with other rewiring approaches.

**Strengths:**

- The overall writing of the paper is relatively clear. However, it is worth noting that the citation format is used incorrectly throughout the entire paper, which significantly affects readability.
- Determining the optimal graph connectivity based on the given node features is inherently a very challenging problem.
- The authors provide experimental results on datasets with large-scale edges.

**Weaknesses:**

- Modeling node feature similarity and label consistency as a concept of “torque” lacks strong justification. In fact, the authors merely intend to express the combined influence of these two metrics on edge probabilities. There are, in practice, many ways to model such combined effects. The choice of using torque for this purpose is not well justified and is therefore unconvincing.
- In line 274, the authors mention selecting the top‑t most similar nodes as candidate nodes. Theoretically, this requires computing the similarity for all $n^2$ pairs of nodes, which results in a computational complexity of $O(n^2)$. Why is this part not considered in the complexity analysis? Additionally, under such complexity, how did the authors conduct experiments on large graphs such as Penn94 and Flickr? Does this involve a mini-batch strategy?
- The authors propose using pseudo-labels to mark unlabeled nodes in order to compute label consistency. This makes the model heavily dependent on the proportion of labeled nodes in the dataset. In their experiments, they use training splits of 48%/32%/20%, but results under lower training proportions should also be provided (e.g., the semi-supervised settings on Cora/CiteSeer/PubMed), especially since most baseline methods do not rely on label information for edge rewiring.
- The Cornell, Texas, and Wisconsin datasets used in the experiments are too small, making the results unreliable. The authors should replace them with larger-scale heterophilic datasets, such as other datasets from the Tolokers and Penn94 series used in the paper.
- Based on the experimental results, THR is actually very close to the baselines, and the performance gains are not significant.

**Questions:**

- Does the experiment on larger-scale datasets involve a mini-batch strategy?

---

### Note · Authors · 2026-01-10

I have read and agree with the venue's withdrawal policy on behalf of myself and my co-authors.